# GIST: Gauge-Invariant Spectral Transformers for Scalable Graph Neural Operators

## Abstract

Adapting transformers to meshes and graph-structured data presents significant computational challenges, particularly when leveraging spectral methods that require eigendecomposition of the graph Laplacian, a process incurring cubic complexity for dense matrices or quadratic complexity for sparse graphs, a cost further compounded by the quadratic complexity of standard self-attention mechanism. Conventional approximate spectral methods compromise the gauge symmetry inherent in spectral basis selection, risking the introduction of spurious features tied to the gauge choice that could undermine generalization. In this paper, we propose a transformer architecture that is able to preserve gauge symmetry through distance-based operations on approximate randomly projected spectral embeddings, achieving linear complexity while maintaining gauge invariance. By integrating this design within a linear transformer framework, we obtain end-to-end memory and computational costs that scale linearly with the number of nodes in the graph. Unlike approximate methods that sacrifice gauge symmetry for computational efficiency, our approach maintains both scalability and the principled inductive biases necessary for effective generalization to unseen graph structures in inductive graph learning tasks. We demonstrate our method's flexibility by benchmarking on standard transductive and inductive node classification tasks, achieving results matching the state-of-the-art on multiple datasets. Furthermore, we demonstrate scalability by deploying our architecture as a discretization-free Neural Operator for large-scale computational fluid dynamics mesh regression, surpassing state-of-the-art performance on aerodynamic coefficient prediction reformulated as a graph node regression task.

## 1 Introduction

Following their incredible success for processing sequential data in Natural Language Processing, Transformers (Vaswani et al., 2017) have been demonstrating a remarkable capacity for handling data of increasing structural complexity. Lee et al. (2019a) have proposed a variant of the transformer block for permutation invariant data with their Set Transformer architecture; Dosovitskiy et al. (2021) have adapted the self-attention mechanism to 2D images with the very influential Vision Transformer architecture; and Bertasius et al. (2021) have extended transformers to video analysis with their Video Vision Transformer (ViViT), demonstrating how attention mechanisms capture both spatial and temporal dependencies across video frames. This progression from sequential text to increasingly structured data indicates a trajectory suggesting that Transformers are poised to tackle even more complex data structures, including irregular meshes and graphs.

Indeed, recent developments in adapting Transformers to graphs have shown promising results in capturing long-range dependencies that traditional Graph Neural Networks (GNNs) struggle with due to their reliance on localized message passing (Dwivedi et al., 2022; Zhu et al., 2023). Unlike GNNs that aggregate information from nearest neighboring nodes by iterating through layers, Transformers can directly capture global relationships across the whole graph through self-attention, enabling them to reason about distant node interactions in a single layer.

**Two Barriers to Scalable Graph Transformers.** However, adapting transformers to graphs introduces two distinct barriers that have not been simultaneously addressed. First, there is a *compu-

*tational barrier*: exact spectral graph embeddings, while theoretically natural, require eigendecomposition of the graph Laplacian, scaling as $\mathcal{O}(N^3)$ for dense graphs or $\mathcal{O}(N^2)$ for sparse graphs (where $N$ is the number of nodes), which is prohibitively expensive for large-scale graphs. Second, there is a *theoretical barrier*: approximate spectral methods that achieve computational efficiency (e.g., random projection-based methods) inadvertently break the gauge invariance of the eigenspace, i.e., the inherent freedom to rotate eigenvectors, flip signs, or choose among degenerate eigenvectors. Importantly, this invariance can also be inadvertedtly broken by the numerics implementing the eigendecomposition of exact spectral methods (Bronstein et al., 2017). This gauge-breaking introduces spurious inductive biases tied to arbitrary numerical choices, causing models trained with one random projection or eigendecomposition to fail catastrophically when evaluated with a different projection or numerical solver, particularly in inductive learning tasks where models must generalize to unseen graph structures.

**The Gauge Invariance Challenge.** Concretely, consider a graph whose spectral embeddings are computed via random projection matrix $R$. A neural network trained on these embeddings $\{R\phi_i\}_i$ will inevitably learn features correlated with the specific choice of $R$. When the same model is evaluated on a different graph, or even re-evaluated on the same graph with a different random projection $R'$ or different numerical eigensolver choices (sign flips, eigenspace ordering, handling of degenerate eigenvalues), the learned features become meaningless. This gauge dependence fundamentally undermines generalization, a critical failure mode for inductive graph learning where models must transfer to unseen structures.

Beyond graph learning, gauge invariance is essential to generate *neural operators with bounded discretization error*. Physical problems (e.g. computational fluid dynamics, structural mechanics, shape analysis) are defined on continuous manifolds but discretized into computational meshes corresponding to graphs. Different mesh resolutions produce different graph Laplacians with different spectral decompositions, each involving arbitrary gauge choices (sign flips, eigenspace rotations, solver artifacts). Without gauge invariance, parameters trained on one discretization fail to transfer to another, and the attention kernels computed from spectral embeddings at different resolutions cannot be compared meaningfully. Gauge invariance ensures that the learned operator converges to the same continuum limit regardless of discretization, enabling provably bounded discretization mismatch error that vanishes as resolution increases.

Existing approaches address only one barrier at a time: (1) spectral methods like SAN (Kreuzer et al., 2021) try to maintain gauge invariance but require full eigendecomposition; (2) approximate spectral methods achieve linear complexity but sacrifice gauge invariance; (3) generic linear transformers reduce attention complexity but ignore graph structure.

**Our Contribution: GIST.** We propose **Gauge-Invariant Spectral Transformers (GIST)**, which overcomes both barriers through a key insight: while random projections break gauge symmetry, the inner products between projected embeddings $\langle R\phi_i, R\phi_j \rangle = \langle \phi_i, (R^T R)\phi_j \rangle \approx \langle \phi_i, \phi_j \rangle$ remain approximately invariant under gauge transformations. By restricting attention operations to only these inner products we recover gauge invariance algorithmically while maintaining $\mathcal{O}(N)$ complexity from random projections and $\mathcal{O}(N)$ from linear attention, achieving end-to-end linear scaling. Our contributions in short are:

1. identifying gauge invariance breaking as a fundamental limitation of approximate spectral methods and characterize when this breaks generalization;

2. proposing GIST, combining gauge-invariant spectral attention with multi-scale linear transformer blocks, achieving theoretical guarantees on complexity and invariance preservation;

3. demonstrating competitive empirical results across diverse settings: transductive (Cora, CiteSeer, PubMed) and inductive (PPI, Elliptic) node classification tasks;

4. establishing GIST as a neural operator with provably bounded discretization mismatch error $\mathcal{O}(n^{-1/(m+4)})$ through gauge invariance, and demonstrating state-of-the-art performance on large-scale mesh regression (DrivAerNet), improving relative L2 error from 20.35% to 20.10% on graphs with ∼500K nodes.

## 2 RELATED WORKS

**Graph Transformers.** Graphormer (Ying et al., 2021) introduces the idea of integrating structural encodings such as shortest path distances and centrality in Transformers. Similarly, Dwivedi et al. (2022) propose LSPE (Learnable Structural and Positional Encodings), an architecture that decouples structural and positional representations. Kreuzer et al. (2021) propose Spectral Attention Network (SAN), which introduces learned positional encodings from the full Laplacian spectrum. Park et al. (2022) develop Graph Relative Positional Encoding (GRPE), which extends relative positional encoding to graphs by considering features representing node-topology and node-edge interactions. Hierarchical Graph Transformer (Zhu et al., 2023) addresses scalability to million-node graphs through graph hierarchies and coarsening techniques.

SpecFormer (Bo et al., 2023) and PolyFormer (Chen et al., 2025b) are recent spectral Transformers that improve over SAN by leveraging approximate spectral bases or low-rank polynomial Laplacian filters to enhance scalability and accuracy on graph tasks.

Several recent architectures aim to improve graph Transformer models via structural encodings and scalable attention. GraphGPS (Rampášek et al., 2022) combines Laplacian or random-walk positional encodings with global attention and local message passing, while Exphormer (Shirzad et al., 2023) replaces full attention with sparse expander-based mechanisms. Tokenized approaches like TokenGT (Hamilton et al., 2017) and NAGphormer (Chen et al., 2023) model graphs as sets of tokens with [CLS]-style readouts. However, these models scale poorly due to full attention complexity, memory-intensive tokenization, and position encoding costs that grow with graph size. As a result, they are rarely evaluated on large inductive node classification tasks such as PPI, Elliptic, or `ogbn-arxiv` due to scalability issues.

**Scalable Attention Architectures.** Recent advances tackled the quadratic scaling of self-attention through various approaches, including cross-attention bottlenecks that map inputs to fixed-size latent representations or concepts (Jaegle et al., 2021b; Rigotti et al., 2022), kernel-based attention mechanisms using random feature approximations (Choromanski et al., 2020), feature map decomposition methods that linearize the attention computation (Katharopoulos et al., 2020), and memory-efficient variants with sub-linear complexity (Likhosherstov et al., 2021). As noted by Dao & Gu (2024), many such linear transformer models are directly related to linear recurrent models such as state-space-models (Gu et al., 2021; 2022; Gu & Dao, 2023; Chennuru Vankadara et al., 2024)

**Neural Operators.** Further addressing the scalability of these graph-based methods is essential for applying them to complex domains such as geometry meshes and point clouds. In these settings, graphs are induced by the connectivity of an underlying continuous object whose discretization is not unique: it can be sampled at arbitrarily many densities and resolutions. High-density discretizations can render the graph prohibitively large, undermining both efficiency and scalability in existing methods. As a result, efficient mesh downsampling and/or re-discretization onto regular lattices (e.g., via SDF-based volumetric grids), and task-aware coarsening learned by GNNs, were commonly required to make these problems tractable.

In recent years, neural operators have shown success in learning maps between continuous function spaces rather than fixed-dimensional vectors. Two properties are crucial here: (i) *discretization invariance*, i.e., a single set of parameters applies across discretizations (meshes, resolutions, and sampling locations) of the same underlying continuum problem; and (ii) *global integration*, i.e., the ability to represent nonlocal interactions via learned integral kernels, rather than being limited to finite-receptive-fields. Formally, a neural operator composes learned integral operators with pointwise nonlinearities, yielding universal approximation results for continuous nonlinear operators and implementations that share weights across resolutions. Our approach preserves these neural operator properties and improves scalability, allowing it to be applied to these cases (Kovachki et al., 2023).

**Foundational operator families.** The Fourier Neural Operator (FNO) parameterizes kernels in the spectral domain and evaluates them with FFT-based spectral convolutions, sharing weights across resolutions and enabling efficient nonlocal interactions on grids (Li et al., 2021). The Graph Neural Operator (GNO) realizes the kernel via message passing, supporting irregular meshes and geometry variation while keeping the learned map discretization-agnostic (Li et al., 2020). Convolutional

Neural Operators (CNOs) define continuous convolutions with learnable kernels and interpolation, specifying the operator in the continuum and discretizing only at runtime (Raonić et al., 2023).

Hybrid designs pair geometry-aware encoders with operator layers to handle complex shapes. GINO couples a graph encoder/decoder with a latent FNO on a proxy grid from SDF or point-cloud inputs and shows convergence across large 3D, multi-geometry problems (Li et al., 2023). Encoder–decoder operator learners, such as DeepONet, use a branch network for inputs and a trunk network for coordinate queries, directly supporting heterogeneous sampling (Lu et al., 2021); U-NO adds a multi-resolution U-shaped backbone for multiscale effects (Rahman et al., 2022).

**Transformers as neural operators.** Self-attention behaves as a learned, data-dependent kernel integral, and with suitable positional features can approximate continuous maps on variable-length sets for discretization-invariant operator learning; cross-attention evaluates outputs at arbitrary coordinates (Tsai et al., 2019; Yun et al., 2020; Lee et al., 2019b; Jaegle et al., 2021a). *Transolver* casts PDE operator learning as attention from query coordinates to context tokens built from input fields, yielding resolution-agnostic inference and strong generalization across meshes (Wu et al., 2024a). Recent operator-oriented transformers, e.g., GNOT, add geometric normalization and gating to stabilize training on irregular meshes and multi-condition PDEs (Hao et al., 2023).

**Positioning GIST.** Existing graph transformers and scalable attention methods address complementary but not simultaneous challenges. Spectral methods like SAN (Kreuzer et al., 2021) leverage the full Laplacian spectrum to maintain theoretical expressiveness but incur significant computational costs from full spectral methods. Approximate spectral methods achieve better scalability but completely forsake gauge invariance, exacerbating generalization failures when arbitrary gauge choice differs between training and testing. Generic linear transformers reduce attention complexity but typically ignore graph structure entirely. GIST uniquely combines graph awareness through spectral embeddings, computational efficiency through random projections, gauge invariance through a modified attention mechanism, and linear attention for end-to-end linear scaling. Furthermore, GIST preserves the discretization-invariance and global integration properties necessary for neural operator applications on mesh regression, unifying graph learning and continuous function approximation in a single framework.

## 3 APPROACH

### 3.1 PRELIMINARIES

**Self-attention and positional encoding.** Given query, key and value representations $q_i$, $k_i$, $v_i$ of $N$ tokens with $i = 1, \ldots, N$, self-attention (Vaswani et al., 2017) famously computes outputs as a weighted sum of values with attention weights determined by query-key similarities:

$$o_i = \sum_{j=1}^{N} \alpha_{ij} v_j, \quad \text{where} \quad \alpha_{ij} = \text{softmax}_j \left( \frac{q_i^\top k_j}{\sqrt{d}} \right). \tag{1}$$

A key insight (Shaw et al., 2018) is that positional information can be injected through relative positional biases: $e_{ij} = \frac{q_i^\top k_j}{\sqrt{d}} + b_{ij}$, where $b_{ij}$ reflects distances between positions. For graphs, this can be generalized by replacing $b_{ij}$ with distance measures reflecting the graph structure as follows.

**Graph Laplacian and spectral embeddings.** The (normalized) *graph Laplacian* $\mathcal{L} = \mathbb{1} - D^{-\frac{1}{2}} A D^{-\frac{1}{2}}$ induces a natural metric via the *resistance distance*: $\Omega(i, j) = (e_i - e_j)^\top \mathcal{L}^\dagger (e_i - e_j)$, where $e_i$ is the $i$th standard basis vector and $\mathcal{L}^\dagger$ is the Moore-Penrose pseudoinverse (Klein & Randić, 1993).

The resistance distance can be expressed via *Laplacian eigenmaps*, which satisfy:

$$\Omega(i, j) = ||\phi_i - \phi_j||^2 \quad \text{where} \quad (\phi_i)_k = \frac{1}{\sqrt{\lambda_k}} (u_k)_i, \tag{2}$$

with $\lambda_k, u_k$ being the eigenvalues and eigenvectors of $\mathcal{L}$. These eigenmaps are natural positional encodings for graphs because their pairwise distances preserve the graph's metric structure (Dwivedi & Bresson, 2021). However, exact computation requires $\mathcal{O}(N^3)$ eigendecomposition, prohibitive for large graphs.

**Approximate spectral embeddings and the gauge invariance problem.** Approximate spectral methods use iterative techniques to compute embeddings with $\mathcal{O}(N \log N)$ complexity. A standard approach is FastRP (Chen et al., 2019), which uses random projections $R \in \mathbb{R}^{r \times N}$ with $r = \mathcal{O}(\log(N)/\epsilon^2)$ to approximate the spectral decomposition via $k$ power iterations on the transition matrix. The resulting approximated eigenmaps are $\tilde{\phi}_i = R\phi_i \in \mathbb{R}^r$.

By the Johnson-Lindenstrauss Lemma, random projections preserve distances approximately: $\Omega(i,j) \approx ||\tilde{\phi}_i - \tilde{\phi}_j||^2 + \mathcal{O}(\epsilon)$, enabling efficient computation. However, the arbitrary choice of the projection matrix $R$ introduces a choice of projection axis, i.e. gauge dependence: models trained on embeddings $\{R\phi_i\}$ will fail when evaluated with a different projection $R'$. Importantly, as noted by Bronstein et al. (2017) also exact spectral embeddings method might accidentally introduce gauge dependence, due to numerical issues or choices introduced by the eigensolver, making them an often overlooked but pervasive concern in the field.

**Motivation for gauge-invariant operations.** While the approximate eigenmaps $\tilde{\phi}_i$ are efficient and preserve distances, as mentioned their gauge dependence is problematic: neural networks trained on these embeddings will learn features correlated with the specific projection matrix $R$. This creates a generalization failure in inductive settings where different graphs or different numerical solvers produce different projections.

Our approach addresses this by using approximate eigenmaps as positional encodings, but restricting the Transformer to operations that depend only on gauge-invariant quantities.

### 3.2 Our Approach: GIST.

The key insight is that while the projection matrix $R$ breaks gauge invariance, the inner products between projected embeddings remain approximately invariant: $(R\phi_i)^\top(R\phi_j) = \phi_i^\top(R^\top R)\phi_j \approx \phi_i^\top \phi_j$. By taking care that all operations depend on the embeddings only through these inner products we preserve gauge invariance by design while maintaining computational efficiency.

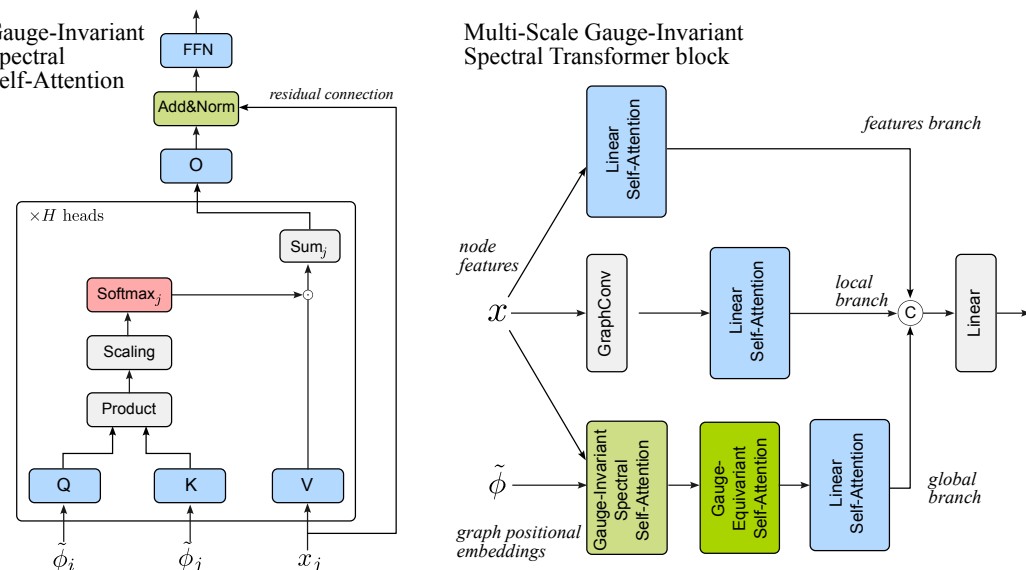

Figure 1: Gauge-Invariant Spectral Transformer. *Left*: Gauge-Invariant Spectral Self-Attention operates on graph positional embeddings $\tilde{\phi}$ as queries and keys, and node features $x$ as values. The output of the self-attention operation is then combined with $x$ through a residual connection. Limiting $\tilde{\phi}$ to queries and keys preserves gauge invariance across the self-attention block. *Right*: Gauge-Invariant Spectral Self-Attention is embedded in a Multi-Scale Gauge-Invariant Spectral Transformer Block which comprises 3 parallel branches inspired by EfficientViT.

**Gauge-Invariant Spectral Self-Attention.** We now introduce our main contribution: **Gauge-Invariant Spectral Transformer (GIST)**. The first ingredient of GIST is *Gauge-Invariant Spectral Self-Attention*, which operates on approximate spectral embeddings $\tilde{\phi}_i = R\phi_i \in \mathbb{R}^r$ but restricts attention computations to *inner products* between embeddings. The key observation is that while the embeddings themselves depend on the arbitrary projection matrix $R$, the inner products $(R\phi_i)^\top(R\phi_j) = \phi_i^\top(R^\top R)\phi_j \approx \phi_i^\top\phi_j$ are approximately gauge-invariant because $R^\top R \approx I$ by Johnson-Lindenstrauss. Thus, attention weights computed from these inner products do not depend on $R$ and generalize across different projections.

Formally, for each node $i = 1, \ldots, N$, Gauge-Invariant Spectral Self-Attention modifies the standard self-attention mechanism as follows:

$$q_i = \tilde{\phi}_i, \quad k_i = \tilde{\phi}_i, \quad v_i = f_v(x_i).$$

This ensures that the attention logits are based on inner products:

$$e_{ij} = \frac{q_i^\top k_j}{\sqrt{d}} = \frac{\tilde{\phi}_i^\top \tilde{\phi}_j}{\sqrt{d}} = \frac{\phi_i^\top R^\top R \phi_j}{\sqrt{d}} \approx \frac{\phi_i^\top \phi_j}{\sqrt{d}},$$

which are approximately gauge-invariant (see Fig. 1, left). By limiting the embeddings to the query-key computation and not using them as values, we ensure that downstream layers (which operate on node features) cannot access gauge-dependent information. Algorithm 1 in Section A.2 explains how we compute graph spectral positional embeddings and Algorithm 2 details the implementation.

**Gauge-Equivariant Spectral Self-Attention.** The Gauge-Invariant Spectral Self-Attention operation thus preserves gauge invariance, but at the cost of giving up a lot of the flexibility of regular self-attention. In particular, there is no mechanism that allows for a modification of the vectors $\tilde{\phi}_i$ through learning. In fact, applying even just a linear operation on $\tilde{\phi}_i$ would break again gauge invariance. However, notice that rescaling each $\tilde{\phi}_i$ by a scalar possibly depending on the node features $s(x_i) \in \mathbb{R}$ would modify the similarity between graph positional embeddings in the same way across gauge choices, since scalars commute with orthogonal projections, meaning that it is an *equivariant* operation across gauges: $(s(x_i)\tilde{\phi}_i)^\top(s(x_j)\tilde{\phi}_j) = s(x_i)s(x_j)(\tilde{\phi}_i^\top \tilde{\phi}_j) = s(x_i)s(x_j)(\phi_i^\top \phi_j)$.

Remarkably, such a gauge-equivariant operation can also be straight-forwardly implemented via a modification of self-attention by modifying equation 1 as follows:

$$q_i = f_q(x_i), \quad k_i = f_k(x_i), \quad v_i = \tilde{\phi}_i,$$

and the output in equation 1 such that it is constrained to operate on $\tilde{\phi}_i$, i.e. $\tilde{\phi}_i^{l+1} = \sum_{j=1}^N \alpha_{ij}v_j$, where $\tilde{\phi}_i^{l+1}$ indicates the graph positional encoding that will be used in the next layer $l + 1$. Algorithm 3 in Section A.2 details how the implementation of Gauge-Equivariant Spectral Self-Attention relates to regular Self-Attention.

**Linear Self-Attention, and Multi-Scale Architecture.** Gauge-Invariant Spectral Self-Attention ensures that we can compute reliable graph positional encoding with linear time complexity in the number of nodes in the graph $N$. In order to maintain that linear scaling end-to-end, the very last component of our architecture aims to address the quadratic scaling of Transformers by implementing a linear version of self-attention. In particular, we implement the *linear transformer* by Katharopoulos et al. (2020). Crucially, as feature map we use $\varphi(x) = \text{ReLU}(x)$, which is a map that induces a kernel $k_0(\cdot)$ corresponding to the arc-cosine kernel (Cho & Saul, 2009). More specifically, for random features $\tilde{\phi}_i, \tilde{\phi}_j \in \mathbb{R}^r$, the attention weights $\langle \varphi(\tilde{\phi}_i), \varphi(\tilde{\phi}_j) \rangle \approx k_0(\tilde{\phi}_i^\top \tilde{\phi}_j)$ converge to a kernel function that depends only on the inner product $\tilde{\phi}_i^\top \tilde{\phi}_j$. Since $\tilde{\phi}_i^\top \tilde{\phi}_j \approx \phi_i^\top \phi_j$ by Johnson-Lindenstrauss (as established earlier), this preserves gauge invariance: attention weights depend only on gauge-invariant inner products between true spectral embeddings. For further considerations on the choice of the feature map $\varphi(\cdot)$ see the note in Appendix A.2.

In order to fully exploit the capabilities of linear attention and mitigate its drawbacks like the reported lack of sharp attention scores compared to softmax attention, we design a parallel architecture inspired from EfficientViT by Cai et al. (2024) who proposed a multi-scale linear attention architecture. Just like EfficientViT our *Multi-Scale Gauge-Invariant Spectral Transformer Block* has 3 parallel branches: a *feature branch* consisting in a linear transformer block acting on node features

$x$ alone, a *local branch* consisting in a graph-convolution layer also acting on $x$ followed by a linear transformer block, and a *global branch* consisting in our Gauge-Invariant Spectral Self-Attention layer followed by a Gauge-Equivariant Spectral Self-Attention layer (which as explained act on both node features $x$ and graph positional embeddings $\tilde{\phi}$) then followed by a linear transformer. In keeping with the analogy with EfficientViT, the role of the graph-convolution layer (which simply averages node features across adjacent nodes) is to emphasize local information, which would be otherwise diffused by linear attention. Conversely, Gauge-Invariant Spectral attention has the role of integrating global information across the graph. This block is represented in the right panel of Fig. 1 and represents a unit layer that is sequentially replicated multiple times.

In Appendix A.5 we report ablation studies empirically showing that all branches meaningfully contribute to the final accuracy of the architecture, specifically for PPI where the full architecture achieves SOTA performance but would not if any of the branches were missing.

**Complexity Analysis.** The computational complexity of GIST is dominated by two components. First, spectral embedding computation via FastRP scales as $\mathcal{O}(N \cdot r \cdot k)$ where $r$ is the embedding dimension and $k$ is the number of power iterations. Second, linear transformer blocks with Gauge-Invariant Spectral Self-Attention on $d$-dimensional node features scale as $\mathcal{O}(N \cdot d^2)$. Overall, this gives end-to-end scaling that is $\mathcal{O}(N \cdot d^2 + N \cdot r \cdot k)$, i.e. linear in the number of nodes $N$. This contrasts with $\mathcal{O}(N^3)$ for exact eigendecomposition and $\mathcal{O}(N^2 d)$ for standard quadratic attention.

## 4 RESULTS

### 4.1 NODE CLASSIFICATION TASKS

To demonstrate the key advantages of GIST, we evaluate it on both transductive and inductive node classification datasets. Transductive tasks are a common graph neural networks paradigm and consist of training and evaluating the model on the same graph, with the goal of predicting at test time node labels that were not provided at training (infilling). Inductive tasks on the other hand, operate on a disjoint set of graphs and aim to predict properties on an entirely new graph.

**Experiment Setup** We evaluate our method on transductive graph benchmarks using the official training, validation, and test splits and evaluation protocols. For each method, we select optimal hyperparameters by optimizing over the validation split of each dataset. GIST specific parameters like the FastRP $k$ (power iterations) and $r$ are treated like regular hyperparameters and are also optimized through HPO. In Appendix A.3 we show that final accuracy is quite robust to variation of these hyperparameters. In addition, as can be expected, larger $r$ tends to result in better accuracy, as it corresponds to better approximation of the original graph Laplacian eigenmaps, and this trend conveniently quickly saturates for relatively low $r$, consistently with the predictions of the Johnson-Lindenstrauss Lemma. To obtain the final result, we conduct a training run on the combined training and validation set and evaluate the model on the corresponding test set. We train across multiple random seeds and report the mean ± standard deviation of the relevant metric.

### 4.1.1 TRANSDUCTIVE TASKS

We evaluate our method on the three standard Planetoid citation benchmarks for the transductive setting where the whole graph is observed at train time: Cora (2,708 nodes, 5,429 edges, 1,433 bag-of-words features, seven classes), CiteSeer (3,327 nodes, 4,732 edges, 3,703 features, six classes), and PubMed (19,717 nodes, 44,338 edges, 500 features, three classes). Train-val-test sets follow the Planetoid public split, and we report node-classification accuracy (Sen et al., 2008; Yang et al., 2016; Kipf & Welling, 2017).

Across these benchmarks, GIST is competitive with strong graph convolutional and transformer-style baselines (see Table 1). On Pubmed, GIST attains the best mean accuracy among the reported methods ($81.20\% \pm 0.41$), narrowly surpassing enhanced GCN variants (e.g., $81.12\% \pm 0.52$) and outperforming GAT/GraphSAGE families. On Cora and Citeseer, GIST achieves results comparable to the top results (within $\sim$1–2 points of GCNII/SGFormer and the enhanced GCN), landing at $84.00\% \pm 0.60$ and $71.31\% \pm 0.50$, respectively.

Table 1: Transductive node classification on the Planetoid benchmarks (Cora, Citeseer, Pubmed). We report test accuracy (%) as mean±std across random seeds using the standard public split (higher is better). Benchmark results are taken from the following references: (Kipf & Welling, 2017; Hu et al., 2021; Luo et al., 2024; Veličković et al., 2018; Chiang et al., 2019; OGB, 2025; Zeng et al., 2020; Chen et al., 2020; Brody et al., 2022; Choi, 2022; Wu et al., 2024b) .

| Model | Cora (Accuracy ↑) | Citeseer (Accuracy ↑) | Pubmed (Accuracy ↑) |
|---|---|---|---|
| GCN (baseline) | $81.60 \pm 0.40$ | $71.80 \pm 0.01$ | $79.50 \pm 0.30$ |
| GraphSAGE | $71.49 \pm 0.27$ | $71.93 \pm 0.85$ | $79.41 \pm 0.53$ |
| GIN | $77.60 \pm 1.10$ | – | – |
| GAT | $83.00 \pm 0.70$ | $69.30 \pm 0.80$ | $78.40 \pm 0.90$ |
| GCNII | $\mathbf{85.50 \pm 0.50}$ | $72.80 \pm 0.60$ | $79.80 \pm 0.30$ |
| GATv2 | $82.90$ | $71.60$ | $78.70$ |
| SGFormer | $84.82 \pm 0.85$ | $72.60 \pm 0.20$ | $80.30 \pm 0.60$ |
| GCN (enhanced) | $85.10 \pm 0.67$ | $\mathbf{73.14 \pm 0.67}$ | $81.12 \pm 0.52$ |
| **GIST (Ours)** | $84.00 \pm 0.60$ | $71.31 \pm 0.50$ | $\mathbf{81.20 \pm 0.41}$ |

### 4.1.2 INDUCTIVE TASKS

We evaluate our method on two inductive benchmarks: PPI, a collection of 24 disjoint tissue-specific protein–protein interaction graphs where nodes (proteins) have 50 features and 121 non–mutually-exclusive GO labels (we use the standard split of 20 graphs for training, 2 for validation, and 2 for testing, and report micro-averaged F1 on the unseen test graphs), and Elliptic, a time-evolving directed Bitcoin transaction graph with 203,769 transactions (nodes), 234,355 payment-flow edges, and 166 features across 49 snapshots, labeled licit/illicit with many nodes unlabeled due to class imbalance (we train on the first 29 time steps, validate on the next 5, and test on the last 14, reporting micro-F1).

On PPI, GIST matches the best large-scale sampling methods and deep residual GCNs (see Table 2), reaching $99.50\% \pm 0.03$ micro-F1, on par with GCNIII and within noise of the strongest GCNII setting ($99.53\%$). On the temporally inductive Elliptic dataset, GIST attains $94.70\% \pm 0.03$ micro-F1. While this trails the strongest GraphSAGE configuration, GIST maintains stable performance across future time steps. These findings collectively demonstrate GIST's effectiveness as a competitive graph learning approach, validating the successful trade-off between computational overhead and representational power. On the Arxiv citation graph (269,343 nodes, 1,166,243 edges, 128 features, and 40 classes) and the Amazon Photo co-purchase network (7,650 nodes, 119,081 edges, 745 features, and 8 classes), we evaluate GIST in the inductive node classification setting following prior Transformer-based benchmarks. On Arxiv, GIST achieves a mean micro-F1 of $72.12\% \pm 0.21$, matching or surpassing several contemporary graph Transformers such as GPS and PolyFormer while maintaining efficient scaling through its linear attention formulation. On the smaller but feature-rich Photo graph, GIST attains $94.42\% \pm 0.40$ micro-F1, competitive with recent spectral and polynomial Transformer variants. These results confirm that GIST preserves accuracy across diverse structural regimes while retaining favorable computational efficiency.

### 4.2 NEURAL OPERATORS

**GIST as Neural Operator.** Many physical problems (CFD, structural mechanics, shape analysis) are defined on continuous manifolds but discretized into computational meshes that form graphs. While message-passing GNNs struggle with long-range interactions and standard transformers incur quadratic complexity on large meshes, GIST's linear scaling enables direct processing of high-resolution meshes with hundreds of thousands of vertices without downsampling. Critically, gauge-invariance is essential for discretization-invariance (the ability key to Neural Operators to apply the same learned parameters across different mesh resolutions): different mesh resolutions produce different spectral decompositions with arbitrary gauge choices (sign flips, rotations, solver artifacts). Without gauge-invariance, parameters trained on one discretization fail to transfer to others.

We now formalize how GIST achieves discretization-invariance through gauge-invariant operations that converge to a continuum operator:

Table 2: Inductive node classification on PPI, Elliptic, Arxiv, and Photo. Results are reported as micro-F1 (higher is better). Benchmark results are taken from the following references: (Chen et al., 2020; Weber et al., 2019; Chiang et al., 2019; Veličković et al., 2018; Zhang et al., 2018; Zeng et al., 2020; Chen et al., 2025b; Brody et al., 2022; Bo et al., 2023)

| Model | PPI (micro-F1 ↑) | Elliptic Bitcoin (micro-F1 ↑) | Arxiv (micro-F1 ↑) | Photo (micro-F1 ↑) |
|---|---|---|---|---|
| GCN (baseline) | $51.50 \pm 0.60$ | 96.10 | $71.74 \pm 0.29$ | $88.26 \pm 0.73$ |
| GraphSAGE | 61.20 | **97.70** | $71.49 \pm 0.27$ | – |
| GAT | $97.30 \pm 0.02$ | 96.90 | – | $90.94 \pm 0.68$ |
| GaAN | 98.70 | – | – | – |
| Cluster-GCN | 99.36 | – | – | – |
| GraphSAINT | 99.50 | – | – | – |
| GCNII | **99.53 $\pm$ 0.01** | – | $72.04 \pm 0.19$ | $89.94 \pm 0.31$ |
| GCNIII | **99.50 $\pm$ 0.03** | – | – | – |
| GATv2 | 96.30 | – | $71.87 \pm 0.25$ | – |
| SpecFormer | 99.50 | – | $72.37 \pm 0.18$ | $95.48 \pm 0.32$ |
| PolyFormer | – | – | **72.42 $\pm$ 0.19** | – |
| Exphormer (LapPE) | | | 72.44 | $91.59 \pm 0.31$ |
| **GIST (Ours)** | **99.50 $\pm$ 0.03** | $94.70 \pm 0.03$ | $72.12 \pm 0.21$ | **94.42 $\pm$ 0.40** |

**Proposition 1.** *Gauge-Invariant Spectral Self-Attention is a discretization-invariant Neural Operator with bounded discretization mismatch error . Let $M$ be a compact $m$-dimensional Riemannian manifold and $\mathcal{G}_n$ a sequence of graphs obtained by sampling $n$ nodes from $M$ with $n \to \infty$. Let $\phi_i^n$ be the Laplacian eigenmaps of $\mathcal{G}_n$ (equation 2). Then:*

*(i) The inner products $\langle \phi_i^n, \phi_j^n \rangle$ converge to the Green's function $G_M(x_i, x_j)$ of the Laplace-Beltrami operator at rate $\mathcal{O}(n^{-1/(m+4)})$.*

*(ii) Gauge-Invariant Spectral Self-Attention is discretization-invariant (can process graphs of arbitrary size), with discretization mismatch error $\mathcal{O}(n^{-1/(m+4)})$ (Gao et al., 2025) for any two discretizations $\mathcal{G}_n$ and $\mathcal{G}_{n'}$ of the same manifold $M$ with $n \le n'$.*

This result follows from three key observations. First, the graph Laplacian converges to the manifold Laplacian (Belkin & Niyogi, 2008; Calder & García Trillos, 2022), so the Laplacian eigenmaps converge to manifold eigenfunctions at rate $\mathcal{O}(n^{-1/(m+4)})$, and their inner products converge to the continuum Green's function. Second, random projections (Johnson-Lindenstrauss) preserve these inner products with controllable error. Third, because the mechanism is gauge-invariant (insensitive to spectral choices like sign flips and eigenspace rotations), the learned parameters do not depend on arbitrary gauge choices in the spectral decomposition. The full proof is deferred to Appendix A.1.

Thus GIST's Gauge-Invariant Spectral Self-Attention component is discretization-invariant while maintaining provable bounds on discretization mismatch error (Gao et al., 2025): the error is dominated by the coarser discretization and vanishes at rate $\mathcal{O}(n^{-1/(m+4)})$ as resolution increases. Since this attention mechanism is a core component of the full GIST architecture, these properties provide theoretical grounding for GIST as a Neural Operator.

**Mesh Based Inductive Task.** To illustrate these properties and the scalability of GIST, we apply it on a real-world continuous mesh based problem: the DrivAerNet dataset. DrivAerNet is a high-fidelity CFD dataset of parametric car geometries comprising 4,000 designs; with each design providing a watertight surface mesh with approximately 0.5M surface vertices per car and accompanying aerodynamic fields (pressure, velocity, wall-shear) plus global coefficients (e.g., $C_d$). We model each car as a graph whose nodes are surface vertices and edges follow mesh connectivity. Our task is *node-level* regression of the surface pressure field on previously unseen cars. Inductive generalization is enforced by holding out entire designs for validation and testing as per the published split. We report per-mesh $R^2$ and RMSE for surface pressure prediction, averaged across test meshes. Among the datasets considered here, DrivAerNet is the largest: both in total data volume and in per-graph node count (Elrefaie et al., 2024).

Table 3: Surface pressure prediction accuracy on **DrivAerNet**. Reported metrics: mean squared error (MSE) and relative $\ell_2$ error (Rel L2). Lower is better for both.

| Year | Model | MSE ($\times 10^{-2}$) | Rel L2 (%) |
|------|-------|------------------------|------------|
| 2024 | RegDGCNN (Elrefaie et al., 2024) | 9.01 | 28.49 |
| 2024 | Transolver (Wu et al., 2024a) | 5.37 | 22.52 |
| 2025 | FigConvNet (Choy et al., 2025) | 4.38 | 20.98 |
| 2025 | TripNet (Chen et al., 2025a) | 4.23 | 20.35 |
| 2025 | **GIST (ours)** | **4.16** | **20.10** |

As illustrated in Table 3, GIST outperforms existing methods on this task. Relevant baseline method MSE and L2 values are pulled from their respective papers. For GIST, 3 layers were used with a hidden dimension of 384 and a node dropout of 0.7. The spectral embedding was computed per vertex with a degree of 96, and the 256 embedding dimensions were appended with the euclidean vertex coordinates and normal vectors. Unlike existing methods on this task, no lossy down-sampling to a lattice grid or arbitrary latent space is required due to the inherent scalability of GIST.

We hypothesize that DrivAerNet's surface meshes have sparse connectivity, intensifying the over-squashing problem in message-passing GNNs where information must propagate through many hops to reach distant vertices. In contrast, GIST's global spectral attention directly captures long-range flow interactions in a single layer, which is critical for accurate surface pressure prediction.

## 5 CONCLUSIONS

We presented GIST, a gauge-invariant spectral transformer architecture that addresses the fundamental computational and theoretical challenges of applying Transformers to graph-structured data. At the core is a simple but powerful insight: while random projections used for computational efficiency break gauge invariance, the inner products between projected embeddings remain approximately invariant. By restricting attention computations to these inner products, we recover gauge invariance algorithmically while maintaining end-to-end linear complexity in the number of nodes.

Our method achieves linear complexity in both memory and computation while preserving gauge invariance, a key property absent from all prior approximate spectral methods. Importantly, we show that gauge invariance can be used as a foundation for bounded discretization mismatch error in Neural Operators: by ensuring that the attention kernel depends only on gauge-invariant inner products that converge to a continuum Green's function, GIST guarantees that learned parameters transfer across mesh resolutions with provable error bounds $\mathcal{O}(n^{-1/(m+4)})$ that vanish as discretization is refined. Unlike approximate methods that sacrifice invariance for efficiency, our approach maintains both scalability and principled inductive biases necessary for effective generalization, with theoretical guarantees that distances are preserved approximately and attention weights are gauge-independent.

Empirically, GIST demonstrates strong performance across diverse settings. On standard graph benchmarks (Cora, CiteSeer, PubMed, PPI, Elliptic), GIST achieves competitive or state-of-the-art results. Most notably, on large-scale mesh regression (DrivAerNet with 500K nodes per graph), GIST achieves 4.16% MSE, improving upon the prior best of 4.23%, and demonstrates that neural operators can be effective without discretization-dependent grid schemes—a practically important finding for scientific computing applications.

By providing a unified framework for graphs and meshes that respects fundamental symmetries while maintaining computational efficiency, GIST opens new directions for foundation models in geometric deep learning and scientific machine learning.

# 6 REPRODUCIBILITY AND ETHICS STATEMENT

To ensure reproducibility of our results we will release our complete source code, including preprocessing scripts, model implementations, and evaluation pipelines, upon publication.

The authors would also like to disclose that a Large Language Model (LLM) was used to minimally aid in the writing of the paper by paraphrasing specific sentences for brevity, clarity, and to avoid stylistic flaws such as repetition. In addition, once a first Related Works section had been compiled by us, and LLM was used to help retrieve and discover possible relevant papers that we had missed. All references provided by the LLM were carefully checked against the literature by the authors.

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

# A APPENDIX

## A.1 PROOF OF PROPOSITION 1: GAUGE-INVARIANT SPECTRAL SELF-ATTENTION

We prove that the Gauge-Invariant Spectral Self-Attention mechanism is discretization-invariant with quantifiable discretization mismatch error through three stages of analysis. The proof establishes that the positional encodings used in the attention mechanism converge to the continuum Green's function, allowing us to bound the discretization mismatch error between any two discretizations of the same manifold.

### A.1.1 STAGE 1: SPECTRAL CONVERGENCE AND GREEN'S FUNCTION

**Proposition 2.** *Let $\mathcal{L}_n$ be the normalized graph Laplacian of a graph $\mathcal{G}_n$ obtained by sampling $n$ nodes from a compact $m$-dimensional Riemannian manifold $M$. Let $\lambda_k^n, u_k^n$ be eigenvalues and eigenvectors of $\mathcal{L}_n$, and $\mu_k, \psi_k$ the eigenvalues and eigenfunctions of the Laplace-Beltrami operator $\Delta_M$ on $M$. Then:*

*(a) Spectral convergence: $|\lambda_k^n - \mu_k| = \mathcal{O}(n^{-1/(m+4)})$ and $\|u_k^n - \psi_k\|_{L^2} = \mathcal{O}(n^{-1/(m+4)})$ (up to log factors).*

*(b) Green's function convergence: The inner products of Laplacian eigenmaps converge to the Green's function:*

$$\langle \phi_i^n, \phi_j^n \rangle = \sum_{\lambda_k^n > 0} \frac{1}{\lambda_k^n} (u_k^n)_i (u_k^n)_j \to G_M(x_i, x_j) + \mathcal{O}(n^{-1/(m+4)}),$$

*where $G_M$ is the Green's function of $\Delta_M$ on $M$.*

*Proof.* Part (a) follows from the spectral convergence theory for graph Laplacians on manifolds (Calder & García Trillos, 2022). With appropriate graph construction, both eigenvalues and eigenvectors of $\mathcal{L}_n$ converge to those of $\Delta_M$ at rate $n^{-1/(m+4)}$ (up to log factors). See Calder & García Trillos (2022) for complete error estimates.

Part (b) follows from part (a) by the spectral theorem. The discrete Green's function (pseudoinverse of the graph Laplacian) is given by the eigenfunction expansion:

$$\langle \phi_i^n, \phi_j^n \rangle = \sum_{\lambda_k^n > 0} \frac{1}{\lambda_k^n} (u_k^n)_i (u_k^n)_j.$$

Using the convergence result from part (a), this sum converges to the continuum Green's function $G_M(x_i, x_j) = \sum_{k=1}^{\infty} \frac{1}{\mu_k} \psi_k(x_i) \psi_k(x_j)$ at rate $\mathcal{O}(n^{-1/(m+4)})$. $\square$

### A.1.2 STAGE 2: RANDOM PROJECTION ERROR

**Proposition 3.** *Let $R \in \mathbb{R}^{r \times N}$ be a random projection with $r = \mathcal{O}(\log(N)/\varepsilon^2)$ constructed via FastRP. For any vectors $v, w \in \mathbb{R}^N$,*

$$\mathbb{P}\left( |\langle Rv, Rw \rangle - \langle v, w \rangle| \leq \varepsilon \|v\| \|w\| \right) \geq 1 - 2e^{-c\varepsilon^2 r}.$$

*Proof.* This follows from the Johnson-Lindenstrauss Lemma (Dasgupta & Gupta, 2003). The key properties:

1. Random projections with $r = \mathcal{O}(\log(N)/\varepsilon^2)$ distort distances by at most a factor $(1 \pm \varepsilon)$ with high probability.

2. For inner products, since distances are preserved, we have $\langle Rv, Rw \rangle = \frac{1}{2}(\|Rv\|^2 + \|Rw\|^2 - \|Rv - Rw\|^2) \approx \frac{1}{2}(\|v\|^2 + \|w\|^2 - \|v - w\|^2) = \langle v, w \rangle$.

3. FastRP specifically uses sparse random matrices that maintain these guarantees while enabling efficient computation (Chen et al., 2019).

See Dasgupta & Gupta (2003) and Chen et al. (2019) for details on FastRP. $\square$

### A.1.3 STAGE 3: GAUGE INVARIANCE

**Proposition 4.** *GIST's learned parameters $\theta$ (projection matrix, transformer weights) do not depend on arbitrary gauge choices (sign flips, rotations) in the spectral decomposition because GIST's computations depend only on gauge-invariant quantities.*

*Proof.* GIST's core attention mechanism is:

$$\alpha_{ij} = \text{softmax}\left(\frac{\langle \tilde{\phi}_i, \tilde{\phi}_j \rangle}{\sqrt{r}}\right), \quad \tilde{\phi}_i = R\phi_i,$$

where $R$ is the random projection matrix. The key insight is that while $\tilde{\phi}_i$ depends on $R$ (the arbitrary gauge choice), the inner products $\langle \tilde{\phi}_i, \tilde{\phi}_j \rangle$ do not (in the limit):

$$\langle \tilde{\phi}_i, \tilde{\phi}_j \rangle = \phi_i^\top (R^\top R)\phi_j \approx \phi_i^\top \phi_j,$$

by Johnson-Lindenstrauss. Thus $\alpha_{ij}$ converges to a gauge-invariant quantity (the continuum Green's function kernel) independent of the choice of $R$.

Since all downstream computations operate on gauge-invariant quantities, the learned parameters $\theta$ do not encode any information about the specific gauge choice in the eigenvector decomposition. $\square$

### A.1.4 DISCRETIZATION MISMATCH ERROR ANALYSIS

Combining all three stages, we establish the discretization mismatch error bound stated in Proposition 1(ii) (Gao et al., 2025). For two discretizations $\mathcal{G}_n$ and $\mathcal{G}_{n'}$ of the same manifold $M$, the attention kernel mismatch is bounded by triangle inequality:

$$\left| \langle \tilde{\phi}_i^n, \tilde{\phi}_j^n \rangle - \langle \tilde{\phi}_i^{n'}, \tilde{\phi}_j^{n'} \rangle \right| \leq \left| \langle \tilde{\phi}_i^n, \tilde{\phi}_j^n \rangle - G_M(x_i, x_j) \right| + \left| G_M(x_i, x_j) - \langle \tilde{\phi}_i^{n'}, \tilde{\phi}_j^{n'} \rangle \right|.$$

Each term on the right decomposes into spectral convergence error (Stage 1) and random projection error (Stage 2), yielding the total bound $\mathcal{O}(n^{-1/(m+4)})$ where $n$ is the coarser discretization. For typical manifold dimensions, random projection errors decay faster than spectral convergence errors, so the latter dominate the discretization mismatch.

### A.2 PSEUDO-CODE

Note that in the pseudocode we use bold notation for matrices and vectors ($\mathbf{A}, \mathbf{\Phi}, \mathbf{Q}$) and follow the row-vector convention standard in machine learning: $\mathbf{\Phi} \in \mathbb{R}^{N \times r}$ has nodes as rows and embedding dimensions as columns. In the main text, we use non-bold notation for compactness, with $\phi_i \in \mathbb{R}^r$ representing individual column vectors and upper case characters denoting matrices.

---

**Algorithm 1** Broken Gauge-Invariance Spectral Embeddings (based on FastRP (Chen et al., 2019))

---

**Require:** Graph adjacency matrix $\mathbf{A} \in \mathbb{R}^{N \times N}$, embedding dimensionality $r$, iteration power $k$
**Ensure:** Matrix of $N$ node graph positional embeddings $\mathbf{\Phi} \in \mathbb{R}^{N \times r}$
  1: Produce very sparse random projection $\mathbf{R} \in \mathbb{R}^{N \times r}$ according to Li et al. (2006)
  2: $\mathbf{P} \leftarrow \mathbf{D}^{-1} \cdot \mathbf{A}$ the random walk transition matrix, where $\mathbf{D}$ is the degree matrix
  3: $\mathbf{\Phi_1} \leftarrow \mathbf{P} \cdot \mathbf{R}$
  4: **for** $i = 2$ to $k$ **do**
  5:     $\mathbf{\Phi}_i \leftarrow \mathbf{P} \cdot \mathbf{\Phi}_{i-1}$
  6: **end for**
  7: $\mathbf{\Phi} = \mathbf{\Phi}_1 + \mathbf{\Phi}_2 + \cdots + \mathbf{\Phi}_k$
  8: **return** $\mathbf{\Phi}$

---

Below we provide pseudo-code for the core computations of GIST, the *Gauge-Invariant Spectral Self-Attention* block and the *Gauge-Equivariant Spectral Self-Attention* block. For illustration purposes, we compare the algorithms to a stripped down implementation of self-attention. We then

---

**Algorithm 2** Gauge-Invariant Spectral Self-Attention (with linear attention)

---

**Require:** Node feature tokens $\mathbf{X} \in \mathbb{R}^{N \times d}$ , graph positional embeddings $\mathbf{\Phi} \in \mathbb{R}^{N \times r}$
**Ensure:** Output sequence $\mathbf{O} \in \mathbb{R}^{N \times d}$ to be applied to features $\mathbf{X}$
1: **// Compute attention matrices**
2: ~~$\mathbf{Q} \leftarrow \mathbf{X} \cdot \mathbf{W}_Q$ where $\mathbf{W}_Q \in \mathbb{R}^{d \times d}$~~    $\mathbf{Q} \leftarrow \mathbf{\Phi}$
3: ~~$\mathbf{K} \leftarrow \mathbf{X} \cdot \mathbf{W}_K$ where $\mathbf{W}_K \in \mathbb{R}^{d \times d}$~~    $\mathbf{K} \leftarrow \mathbf{\Phi}$
4: $\mathbf{V} \leftarrow \mathbf{X} \cdot \mathbf{W}_V$ where $\mathbf{W}_V \in \mathbb{R}^{d \times d}$
5: **// Compute linear attention with feature map $\varphi(x) = \mathrm{ReLU}(x)$**
6: $\tilde{\mathbf{Q}}, \tilde{\mathbf{K}} \leftarrow \varphi(\mathbf{Q}), \varphi(\mathbf{K})$
7: $\mathbf{S} \leftarrow \tilde{\mathbf{K}}^T \mathbf{V}$    ▷ Compute key-value matrix: $\mathbb{R}^{r \times d}$
8: $\mathbf{Z} \leftarrow 1/(\tilde{\mathbf{Q}}(\tilde{\mathbf{K}}^T \mathbf{1}_N) + \epsilon)$    ▷ Normalization factors: $\mathbb{R}^N$
9: $\mathbf{O} \leftarrow (\tilde{\mathbf{Q}}\mathbf{S}) \odot \mathbf{Z}$    ▷ Normalized output: element-wise product
10: **return** $\mathbf{O}$

---

**Algorithm 3** Gauge-Equivariant Spectral Self-Attention (with linear attention)

---

**Require:** Node feature tokens $\mathbf{X} \in \mathbb{R}^{N \times d}$ , graph positional embeddings $\mathbf{\Phi} \in \mathbb{R}^{N \times r}$
**Ensure:** Output sequence $\mathbf{O} \in \mathbb{R}^{N \times d}$ to be applied to graph positional embeddings $\mathbf{\Phi}$
1: **// Compute attention matrices**
2: $\mathbf{Q} \leftarrow \mathbf{X} \cdot \mathbf{W}_Q$ where $\mathbf{W}_Q \in \mathbb{R}^{d \times d}$
3: $\mathbf{K} \leftarrow \mathbf{X} \cdot \mathbf{W}_K$ where $\mathbf{W}_K \in \mathbb{R}^{d \times d}$
4: ~~$\mathbf{V} \leftarrow \mathbf{X} \cdot \mathbf{W}_V$ where $\mathbf{W}_V \in \mathbb{R}^{d \times d}$~~    $\mathbf{V} \leftarrow \mathbf{\Phi}$
5: **// Compute linear attention (Katharopoulos et al., 2020)**
6: $\tilde{\mathbf{Q}}, \tilde{\mathbf{K}} \leftarrow \varphi(\mathbf{Q}), \varphi(\mathbf{K})$
7: $\mathbf{S} \leftarrow \tilde{\mathbf{K}}^T \mathbf{V}$    ▷ Compute key-value matrix: $\mathbb{R}^{d \times r}$
8: $\mathbf{Z} \leftarrow 1/(\tilde{\mathbf{Q}}(\tilde{\mathbf{K}}^T \mathbf{1}_N) + \epsilon)$    ▷ Normalization factors: $\mathbb{R}^N$
9: $\mathbf{O} \leftarrow (\tilde{\mathbf{Q}}\mathbf{S}) \odot \mathbf{Z}$    ▷ Normalized output: element-wise product
10: **return** $\mathbf{O}$

---

point out the modifications that our algorithms apply to that basic functionality by indicating in red any addition to vanilla self-attention and in strike-through text anything that has to be removed.

**Note on the choice of Feature Map $\varphi$.** While in Algorithm 3 we do not need to impose that restriction, in Algorithm 2 we use the feature map $\varphi(x) = \mathrm{ReLU}(x)$. This choice is theoretically motivated: when applied element-wise to random features, ReLU induces the *arc-cosine kernel* (Cho & Saul, 2009). Specifically, for vectors $\mathbf{a}, \mathbf{b} \in \mathbb{R}^r$, the inner product $\langle \varphi(\mathbf{a}), \varphi(\mathbf{b}) \rangle$ converges (as $r \to \infty$) to a kernel function $k_0(\mathbf{a}^\top \mathbf{b})$ that depends only on the inner product $\mathbf{a}^\top \mathbf{b}$.

This property is crucial for preserving gauge invariance: since the attention weights are computed as $\langle \varphi(\tilde{\phi}_i), \varphi(\tilde{\phi}_j) \rangle \approx k_0(\tilde{\phi}_i^\top \tilde{\phi}_j)$ and $\tilde{\phi}_i^\top \tilde{\phi}_j \approx \phi_i^\top \phi_j$ by Johnson-Lindenstrauss (as established in Section A.1), the resulting attention pattern depends only on gauge-invariant inner products between the spectral embeddings.

We note that the original linear attention work by Katharopoulos et al. (2020) used $\varphi(x) = \mathrm{elu}(x) + 1$. Empirically, this feature map also tends to work well in practice, and it is similar to ReLU in producing non-negative outputs. However, it is not known to correspond to any particular kernel function, and thus the theoretical guarantee of gauge invariance via kernel structure does not apply. Investigating other feature maps corresponding to different kernel functions (e.g., polynomial kernels, random Fourier features for RBF-like kernels) is left for future work.

## A.3 GIST HYPERPARAMETERS ROBUSTNESS

In order to study the sensitivity of GIST's performance to variations in its spectral embedding hyperparameters, we train multiple simplified GIST architectures (two-block Gauge-Invariant Spectral Self-Attention linear transformers) on the Cora benchmark while varying the power iteration pa-

rameter $k$ and the embedding dimension $r$ in the FastRP approximation. These parameters directly control the quality of spectral embeddings while balancing computational efficiency.

As shown in Figure 2, GIST exhibits robust performance across a wide range of both parameters. The left panel sweeps $k$ with $r$ fixed, a clear but relatively shallow peak in accuracy around the optimal value of $k \approx 32$. This on one hand suggests that even modest iteration counts are sufficient to capture the essential spectral structure, but also indicates that the choice of the power iteration is quite robust. The right panel varies embedding dimension $r$ with $k$ fixed, demonstrating a smooth monotonic improvement as $r$ increases. Crucially, saturation occurs relatively quickly: performance gains beyond $r = 256$ are marginal, validating our choice of reasonable embedding dimensions that maintain computational efficiency.

These results empirically validate two important properties: (1) GIST does not require extensive hyperparameter tuning around these spectral parameters, suggesting stable generalization; and (2) the linear end-to-end complexity achieved with modest $k$ and $r$ values is both computationally practical and empirically effective. Combined with the gauge-invariance guarantees that prevent dependence on arbitrary spectral choices, these hyperparameters provide a principled way and empirically robust way to control the approximation quality of spectral embeddings without sacrificing scalability.

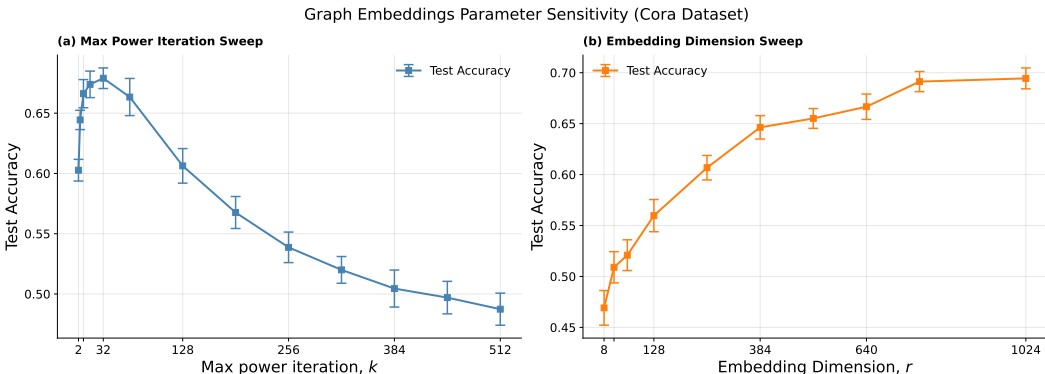

Figure 2: Sensitivity study of GIST spectral embeddings parameters. The plots show the final test accuracy of a two-block Gauge-Invariant Spectral Self-Attention linear transformer trained on Cora while sweeping over the power iteration parameter $k$ with $r = 256$ (left panel), and sweeping over the embedding dimension $r$ with $k = 32$ (right panel). Test accuracy is fairly robust around the best value of either parameter. As expected, $r$ is monotonically related to higher performance, as higher $r$ correspond to better approximations of the eigenmaps. Accuracy conveniently saturate relatively fast, justifying the use of reasonably low $r$. The plots show mean test accuracy averaged across 10 seeds and corresponding standard deviation as error bars.

## A.4 GIST Scalability Study

From a computational standpoint, the end-to-end cost of GIST is dominated by two components: spectral embedding generation and the subsequent transformer blocks. For the former, we employ a FastRP-style approximation in which the Laplacian spectral information is captured via repeated multiplication of a sparse random walk matrix with a low-dimensional random projection. Each power iteration requires $\mathcal{O}(|E|r)$ operations, where $|E|$ is the number of edges and $r$ is the embedding dimension, and the total cost over $k$ iterations is $\mathcal{O}(k|E|r)$. On meshes and graphs with bounded average degree, $|E| = \mathcal{O}(N)$, so the overall spectral embedding stage scales linearly in the number of nodes $N$. This embedding is computed once per graph and then reused across all GIST layers, so its cost is amortized over the full network depth.

The GIST layers themselves preserve this linear scaling. Each block combines: (i) a feature branch based on linear attention, (ii) a local branch using graph convolution followed by linear attention, and (iii) a global branch using Gauge-Invariant and Gauge-Equivariant Spectral Self-Attention followed by linear attention. In all cases, attention is implemented in the form $\varphi(Q)(\varphi(K)^\top V)$ with an element-wise feature map $\varphi(\cdot)$, which yields $\mathcal{O}(Nd^2)$ complexity for $d$-dimensional features in-

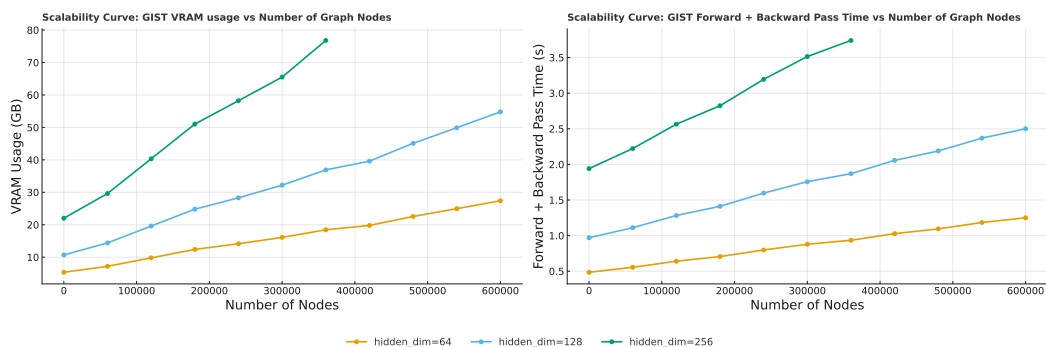

Figure 3: Scalability study of GIST. All experiments use a fixed 3-layer model while varying the hidden dimensionality. VRAM consumption was measured as a function of the number of nodes in the input graph. Graph sizes were controlled using random node dropout applied to samples from the Drivaernet dataset, enabling a systematic evaluation of memory scaling behavior.

stead of the $\mathcal{O}(N^2 d)$ cost of quadratic attention. Together with the $\mathcal{O}(N)$ spectral embedding stage, this results in an overall complexity of $\mathcal{O}(N(d^2 + rk))$ per forward pass, i.e., linear in the number of nodes. The empirical VRAM and wall-clock measurements in Figure 3 corroborate this analysis: both memory usage and forward time grow approximately linearly with the number of graph nodes for all hidden dimensions, up to graphs with hundreds of thousands of nodes sampled from DrivAerNet.

## A.5 MULTI-SCALE ARCHITECTURE ABLATION STUDY

To validate the design choices in the Multi-Scale GIST architecture, we systematically ablate each of the three parallel branches shown in Figure 1 (right panel) on the PPI dataset, where the full architecture achieves state-of-the-art performance (see Table 2). For each ablation, we train with identical hyperparameters but remove one branch: (1) feature processing, (2) local graph convolution, or (3) global spectral attention. All experiments are repeated across 20 random seeds.

Table 4 shows that all three branches contribute meaningfully, with performance drops ranging from 4.29% to 7.90%. The local branch (Branch 2) has the strongest impact ($-7.90\%$), validating the EfficientViT-inspired design principle that local operations provide focused information complementing the diffuse patterns from linear attention. The global spectral branch (Branch 3, $-4.29\%$) confirms that long-range dependencies are essential, while the feature branch (Branch 1, $-5.04\%$) provides complementary signal beyond structural information. Overall, these results demonstrate that the Multi-Scale GIST effectively integrates complementary information sources.

Table 4: Ablation study on PPI dataset showing the contribution of each branch of the Multi-Scale GIST (see Figure 1, right panel). Test accuracy is reported as percentage of baseline performance. Results averaged over 10 seeds with standard deviations indicated as uncertainty intervals.

| Ablation | Test Accuracy (% baseline) | $\Delta$ (%) |
|---|---|---|
| No ablation | 100.0 | 0.0 |
| Branch 1 (feature) | $95.0 \pm 2.8$ | $-5.0$ |
| Branch 2 (local) | $92.1 \pm 1.8$ | $-7.9$ |
| Branch 3 (global) | $95.7 \pm 3.7$ | $-4.3$ |

