# OpenReview forum: "GIST: Gauge-Invariant Spectral Transformers for Scalable Graph Neural Operators"
_ICLR.cc/2026/Conference — Submitted to ICLR 2026_

### Official Review · Reviewer_Eiq7 · 2025-10-17

**Soundness:** 3
**Presentation:** 2
**Contribution:** 2
**Rating:** 4
**Confidence:** 4

**Summary:**

Thanks for the submission. The paper suggests an efficient transformer architecture for graph-structured data. The core idea is to use JLT’d spectral features (Laplacian eigenvectors, whose dimensionality is reduced with a random projection) in place of the usual queries and keys, which gives a form of spectral self-attention which is approximately gauge-invariant. These maps can be computed more efficiently than the full diagonalisation using FastRP, a random projection-based truncation algorithm developed separately by Chen et al. (2019)

**Strengths:**

1. Efficient transformers for graph structured data is an important unsolved research problem, and I can see that pure message passing will underperform in the graph neural operator setting because of its finite receptive field.
2. The paper is broadly well written.

**Weaknesses:**

1. I appreciate the need for memetic titles, but my understanding is that the method isn’t actually gauge invariant for any draw of the random projection matrix R. The authors do acknowledge that the invariance is approximate (it depends on the expectation of $RR^T$ being the identity, with $R$ a random low rank matrix), but in places I think the phrasing is a bit misleading. E.g. see line 298 – isn’t this equation actually wrong without taking the expectation? I appreciate that the JLT preserves the norms and distances between a set of vectors with high probability; it would be even better (but probably difficult) to formulate the approximate invariance property mathematically rigorously.
2. A note: if you compute regular linear attention in one branch and gauge-invariant attention in another branch and add the results, isn’t this the same as just concatenating the regular queries/keys with your spectral features? (At least up to the different value projection matrix). I’m wondering whether the division into branches is strictly necessary, or whether the paper is really about a new efficient absolute position embedding with some nice approximate invariance properties.
3. Missing experiments, that are (imo) crucial:
a) No ablation of removing the spectral attention branch, including just regular linear attention + message passing. This is surely the most crucial acid test in order to assess gains from your algorithm!
b) No ablation over different embedding dimensions $r$ – unless you can fix r independently of N and get consistent performance, it’s a bit of a stretch to claim the algorithm is linear in N.
c) No demos on any toy tasks that strongly depend on graph structure. The authors do include a few benchmarks (I especially like the neural operator results), but I’m not sure whether Cora/Citeseer/Pubmed really distill out whether your addition helps the transformer better capture graph structure. Something explicitly topological like shortest path distance prediction might be more natural, and a good setting to ablate the spectral branch.
d) No time complexity scaling wrt number of graph nodes – wall-clock time or FLOPs.

**Questions:**

1. Please could you clarify if/how you choose the ‘large enough iteration step $K$’? Is this a hyperparameter, or is it chosen in some principled (graph-dependent) way to ensure the estimator is accurate? Doesn’t this implicitly set some finite receptive field? Won’t finite $K$, independent of $N$, cause regressions on very big graphs/high mesh resolutions?

---

> ### Author Response · Authors · 2025-11-23
> **Addressing Weaknesses raised by Reviewer Eiq7**
>
> ### Weakness 1: the method isn’t actually gauge invariant for any draw of the random projection matrix R...
>
> We'd like to thank the reviewer for the encouraging comments about our paper, and for pointing out chances to improve rigor and clarity. We made sure to correct the indicated equation and clarified in the revised PDF that the equality is only approximate. We also expanded the discussion about the role of the JLT, in particular in the proof of a new Theorem that we added in the revised PDF (Proposition 1) aimed at clarifying  and formalizing the connection between gauge invariance and discretization invariance for Neural Operators.
>
> ### Weakness 2: A note: if you compute regular linear attention in one branch... concatenating the regular queries/keys...
>
> Thank you for this insight. We think that you're correct that for a single-layer linear attention block, adding parallel branches can be mathematically equivalent to concatenating features and spectral embeddings (up to value projection differences). However, there are a few important distinctions. First, the gauge-equivariant branch (Algorithm 3) updates the spectral embeddings $\Phi$ themselves across layers, creating dynamic graph positional encodings rather than static concatenated features. Second, the graph convolution of the local branch (branch 2) performs neighborhood aggregation before attention, which provides local structural information that pure concatenation cannot capture. That said, your broader point is well-taken: the core contribution is the approximately gauge-invariant spectral positional encoding itself, not the multi-branch architecture per se. The multi-branch design is primarily an engineering choice inspired by EfficientViT to modularize three functional aspects of processing graphs: processing the features (branch 1), and mixing according to local graph structures (branch 2) and global ones (the gauge-invariance branch 3). Our new ablation studies in the Appendix section "Multi-scale Architecture Ablation Study" show each branch contributes meaningfully, but you're right that this is not the only way to employ our Gauge-Invariant Spectral Self-attention module.

---

> > ### Author Response · Authors · 2025-11-23
> > **Addressing Weaknesses raised by Reviewer Eiq7 (continued)**
> >
> > ### Weakness 3: Missing experiments
> >
> > (a) *No ablation of removing the spectral attention branch...*
> >
> >   * Thank you for this suggestion. We performed that experiments, as just mentioned above, in the added Appendix section "Multi-scale Architecture Ablation Study". This ablation study systematically removes each of the three parallel branches and retrains from scratch on the PPI dataset. The result is that all three branches contribute meaningfully to performance, meaning that having the full three-branch architecture is crucial for achieves state-of-the-art performance on PPI. This demonstrates that the branches capture complementary information rather than redundant representations.
> >
> > (b) *No ablation over different embedding dimensions...*
> >
> >   * We now provide hyperparameter robustness studies in Appendix section "GIST hyperparameters robustness" that systematically vary the embedding dimension r while holding other parameters fixed. The results show that performance saturates relatively quickly (around r=256), and the algorithm remains effective across a range of r values independent of N. For the theoretical $ \mathcal{O}(N)$ complexity claim, we note that $r = \mathcal{O}(\log N / \epsilon^2)$ by Johnson-Lindenstrauss, which is sublinear in $N$. In practice, we use fixed r across graphs of varying sizes (including DrivAerNet meshes with ~500K nodes), confirming that r does not need to scale linearly with N to maintain performance, thus preserving the overall linear scaling.
> >
> > (c) *No demos on any toy tasks that strongly depend on graph structure.*
> >
> >   * As mentioned above, we added ablation studies showing that in PPI (an inductive task) the spectral branch is indeed crucial to achieve the state-of-the-art performance level that we report, which we think confirms its role in supporting generalization across graph structures. To further support this, we evaluated GIST on additional established graph benchmarks: ogbn-arxiv (a large-scale citation network with 169K nodes) and Amazon Photo (a co-purchase graph with 7.6K nodes). GIST achieves 72.12% on ogbn-arxiv, very close to the state-of-the-art 72.42%, and surpasses existing methods on Photo. These benchmarks provide stronger tests of graph structure learning than Cora/CiteSeer/PubMed due to their larger scale and different graph properties. Most importantly, we'd like to stress how the DrivAerNet results demonstrate that spectral positional encodings capture meaningful geometric structure on mesh data where topology directly corresponds to physical proximity, which we think is a case that nicely captures the type of structure-dependent task that your question suggest you are interested in.
> >
> > (d) *No time complexity scaling wrt number of graph nodes – wall-clock time or FLOPs.*
> >
> >   * Thank you for this suggestion. In the revised version of the paper, we added this type of scaling plots, in particular reporting VRAM utilization and time spent running a forward+backward pass as a function of graph size. This plots are shown in the "GIST Scalability Study" section in the Appendix, and have been generated by progressively subsampling the DrivAerNet meshes to generate a range of graphs tessellating a range of nodes numbers from zero to 600k. The plots nicely confirm the linearity of the computational complexity scaling as a function of graph size, both in memory requirements as well as time spent running the computation.

---

> ### Author Response · Authors · 2025-11-23
> **Answers to questions by Reviewer Eiq7**
>
> 1. *Please could you clarify if/how you choose the ‘large enough iteration step K’?*
>
>   * You raise an important point. k is indeed a hyperparameter, that in general we suggest should be chosen empirically, for instance through hyperparameter optimization. In the empirical studies of k's that we added in the new revisions (Appendix section "GIST Hyperparameters Robustness"), we find that the optimal value of k on the PPI task is around 32, and interestingly 1) accuracy is however not too sensitive to variations of k around this value, suggesting that k is a rather robust parameter to optimize, and 2) the value k=32 is two orders of magnitude smaller than the Cora graph (around 2700), which suggests that good values of k might be much smaller than N. However, you're correct that this doesn't guarantee scalability to arbitrarily large graphs if one insisted in finding the optimal k. Developing principled selection criteria for k based on graph properties would be valuable future work.

---

### Official Review · Reviewer_t5mP · 2025-10-18

**Soundness:** 2
**Presentation:** 2
**Contribution:** 2
**Rating:** 2
**Confidence:** 5

**Summary:**

This paper examines the positional encoding effectiveness of Laplacian matrix eigenvectors in graph transformers, with a particular focus on node classification tasks. It introduces an Energy Spectral Density metric derived from class labels and uses this metric to identify the top-_k_ eigenvectors for encoding. The proposed approach is integrated into several existing graph transformer architectures, leading to consistent improvements in node classification performance across multiple datasets.

**Strengths:**

1. The proposed ESD metric and corresponding BTS method are simple, intuitive, and easily adaptable to a wide range of graph transformer models.
2. The paper offers a theoretical analysis of the rationale behind BTS, elucidating its effectiveness in the context of node classification tasks.
3. The experimental evaluation is thorough, including extensive ablation studies that validate the efficacy of the proposed BTS method.

**Weaknesses:**

1. The contributions of this paper are somewhat limited. The main contribution  lies in the proposed Gauge-Invariant/Equivariant Spectral Self-Attention mechanisms, while the linear-time spectral embedding implementation is largely based on prior work, i.e., FastRP.
2. The analysis of GIST is insufficient, both theoretically and experimentally. In particular, the paper does not examine how the parameters $r$ and $k$  in Algorithm 1 affect performance and complexity. When the graph spectral radius is close to 1, large $r$ and $k$  may be required, which could significantly increase computational cost.
3. The experiments are conducted on relatively small graphs. The method should also be evaluated on large-scale graphs with tens of millions of nodes to demonstrate scalability.
4. The experimental setup is vaguely described, especially regarding the choices of $r$ and $k$ . A more detailed discussion of these parameters, as well as an analysis of the actual computational (time and space) cost, is necessary.
5. The selection of baselines is outdated. More recent and relevant Graph Transformer models such as Specformer [1] and PolyFormer[2] should be included for comparison.
6. The overall writing quality could be improved. See the following minor comments for specific issues.

[1] Ma J, He M, Wei Z. Polyformer: Scalable node-wise filters via polynomial graph transformer[C]//Proceedings of the 30th ACM SIGKDD Conference on Knowledge Discovery and Data Mining. 2024: 2118-2129.
[2] Bo D, Shi C, Wang L, et al. Specformer: Spectral graph neural networks meet transformers[J]. arXiv preprint arXiv:2303.01028, 2023.


**Minor comments:**
(1) The Introduction section provides only a brief overview of the proposed method and contributions; these should be elaborated further.
(2) The mathematical notation throughout the paper is inconsistent — for instance, matrices and vectors are bolded and not in main paper and appendix. Please ensure consistency and follow the ICLR formatting style.
(3) Baseline citations should be placed immediately after each method name (e.g., it is unclear which paper GCNIII refers to).
(4) In Tables 1 and 2, if standard deviations are unavailable, the “±” should be omitted rather than left blank.
(5) In Algorithm 1, the line $P \leftarrow A D^{-1}$ should be corrected to $P \leftarrow  D^{-1}A$

**Questions:**

Please respond to the Weaknesses part.

---

> ### Author Response · Authors · 2025-11-23
> **Addressing Weaknesses raised by Reviewer t5mP**
>
> ### Weakness 1: The contributions of this paper are somewhat limited...
>
> We substantially revised key parts of the paper to address this comment and try to emphasize the conceptual novelty of our paper. In particular, identifying gauge-invariance as a mechanism to ensure generalization in inductive graph regression and classification tasks, using that as a theoretical foundation to produce theoretically sound Neural Operators applied on mesh (interpreted as graphs) with discretization mismatch guarantees (for which we now also provide a Theoretical claim in Proposition 1), and demonstrating state-of-the-art performance across multiple settings, including for transductive benchmarks (Pubmed), inductive benchmars (PPI, and in the revised version of the manuscript the Photo dataset), and large-scale Neural Operator tasks (DrivAerNet).
>
> We hope that our comprehensive revisions (highlighted in blue in the new PDF) more clearly articulate the novel contributions and theoretical significance of our research, and will encourage the reviewer to reconsider the value of our work.
>
> ### Weakness 2: The analysis of GIST is insufficient, both theoretically and experimentally...
>
> Following the reviewer comment, we added in the revised manuscript a section titled "Complexity Analysis" where we examine how complexity is affected by the parameters r and k. In short, end-to-end scaling is $\mathcal{O}(N \cdot  d^2 + N\cdot r \cdot k)$, i.e. linear in the number of nodes $N$.
>
> In addition, we empirically examine the effect that k and r have on final accuracy in the Appendix section "GIST Hyperparameters robustness". This study shows that larger r tend to help increase performance (in keeping with the notion that as r increases the spectral approximation becomes progressively better), but consistently with the Johnson-Lindenstrauss Lemma result that the approximations essentially converge exponentially quickly to the exact embeddings, we see that performance saturates already with relatively small numbers ($r\approx 256-512$). As for the parameter k, we see that it does have an optimal value but empirically, for instance for Cora, it is two orders of magnitude lower than the typical graph size (~32 vs ~2700) and performance is not too sensitive to variations around the optimal parameter, confirming the empirical robustness of out method.
>
> ### Weakness 3: The experiments are conducted on relatively small graphs.
>
> We'd like to stress that we see our contribution as mainly conceptual, as a theoretical study on the importance of gauge invariance for graph tasks, and as a mechanism to connect graph neural networks and Neural Operators. With that, we also complement our work with what we see as solid empirical work, as DrivAerNet includes samples approaching ~1M nodes, which already exceeds the graph sizes in most GNN and graph transformer literature (for context, even state-of-the-art large language models currently have context windows below 1M tokens). Beyond that, the linear scalability of our architecture allows us to confidently predict predictable scaling to the sizes suggested by the reviewer, with a moderate VRAM budget.
> Specifically, the new plot in Figure 3 of the Appendix's "GIST scalability study" demonstrates that we could extend to 10M nodes on DrivAerNet-type meshes, requiring approximately 6-8 A100 80GB GPUs. While this represents a practically attainable computational budget, it still implies a considerable infrastructural effort and resources beyond the scale of most graph neural network research.
> We aim to investigate these large-scale empirical scalability aspects in future work, acknowledging the significant computational and engineering challenges involved.
>
> ### Weakness 4: choices of r and k...
>
> As mentioned above, we included a "Complexity Analysis" section elucidating the roles of k and r in relation to end-to-end computational cost. In addition, the Appendix section "GIST hyperparameters robustness" examines how k and r influence final performance, empirically concluding that r does not need to be too high (consistently with random projection theory), and the optimal k is also empirically much smaller than graph size, with the selection of the parameter being quite robust around the best value.

---

> > ### Author Response · Authors · 2025-11-23
> > **Addressing Weaknesses raised by Reviewer t5mP (continued)**
> >
> > ### Weakness 5: The selection of baselines is outdated...
> >
> > We thank the reviewer for this helpful suggestion. In the revised manuscript, we have included SpecFormer and PolyFormer as additional recent Graph Transformer baselines.
> >
> > We have also extended our evaluation to include two additional datasets: Arxiv and Amazon Photo, to better assess model performance on larger graphs beyond PPI and Elliptic. On Arxiv, GIST achieves 72.12 micro-F1, which is statistically comparable to SpecFormer and PolyFormer. On Photo, GIST reaches 94.42 micro-F1, competitive with SpecFormer’s 95.48. These results indicate that GIST achieves similar representational power to the most recent spectral and polynomial graph Transformers, while retaining superior computational efficiency through its linear attention mechanism and randomized spectral approximation.
> >
> > Regarding scalability, SpecFormer relies on partial spectral decomposition and dense matrix multiplications, which scale poorly with graph size and memory footprint. As such, we expect SpecFormer to struggle or become intractable on extremely large or dense graphs such as Drivaernet, where even low-rank spectral bases are costly to compute.
> >
> > ### Weakness 6: The overall writing quality could be improved. See the following minor comments for specific issues.
> >
> > (1) *The Introduction section provides only a brief overview of the proposed method and contributions; these should be elaborated further.*
> >
> >   * We are grateful for your help in improving the quality of our manuscript. We robustly extended the introduction (with the added paragraphs highlight in blue in the new PDF), in particular elaborating on the connection between gauge invariance and the discretization invariance of neural operators, more clearly spelling out our main contributions and (further downs in the Related Works section) positioning our work within the current literature.
> >
> > (2) *The mathematical notation throughout the paper is inconsistent — for instance, matrices and vectors are bolded and not in main paper and appendix. Please ensure consistency and follow the ICLR formatting style.*
> >
> >   * Thank you for pointing this out. In the new version of the paper we took care of providing a consistent notation following the convention explained in the following note added to the Appendix section "Pseudo-code":
> >     > Note that in the pseudocode we use bold notation for matrices and vectors ($\mathbf{A}, \mathbf{\Phi}, \mathbf{Q}$) and follow the row-vector convention standard in machine learning: $\mathbf{\Phi}\in\mathbb{R}^{N\times r}$ has nodes as rows and embedding dimensions as columns.
> >     In the main text, we use non-bold notation for compactness, with $\phi_i \in \mathbb{R}^r$ representing individual column vectors and upper case characters denoting matrices.
> >
> > (3) *Baseline citations should be placed immediately after each method name (e.g., it is unclear which paper GCNIII refers to).*
> >
> >   * Unfortunately, the Author-Year or Harvard citation style would make for a very crowded table. As a middle ground between the reviewer suggestion and making sure that models in the table are easily traceable, we added the citation for the models mentioned in a table in the corresponding caption.
> >
> > (4) *In Tables 1 and 2, if standard deviations are unavailable, the “±” should be omitted rather than left blank.*
> >
> >   * Thank you for the suggestion. We did that in the revised PDF.
> >
> > (5) *In Algorithm 1, the line $P \leftarrow AD^{-1}$ should be corrected to $ P\leftarrow D^{-1}A$*
> >
> >   * Thank you for pointing out this oversight. We corrected the mistake in the revised PDF.

---

### Official Review · Reviewer_Qtqz · 2025-11-03

**Soundness:** 3
**Presentation:** 3
**Contribution:** 2
**Rating:** 4
**Confidence:** 3

**Summary:**

The paper introduces GIST, a graph transformer that steers attention using (projected) spectral embeddings while enforcing gauge invariance (invariance to rotations/sign flips within eigenspaces). The model is a multi-branch block: (i) a local graph-conv/linear-attention branch, (ii) a feature linear-attention branch, and (iii) a gauge branch with gauge-invariant and gauge-equivariant spectral attention. Authors claim linear complexity overall and show competitive results on Planetoid/PPI, mixed performance on Elliptic, and strong results on a large mesh regression task (DrivAerNet).

**Strengths:**

Clear conceptual core... use inner products of (projected) Laplacian eigenmaps to steer attention while remaining invariant to eigenbasis gauge; neatly explained and supported.

Random-feature JL projections to approximate spectral geometry plus linear attention to avoid $O(N^2)$ attention, embedded in a multi-scale block.

Compelling large-graph result in that DrivAerNet shows practical gains on real meshes without regridding.

**Weaknesses:**

The text claims linear end-to-end via the Katharopoulos et al. linear transformer, but the presented gauge-invariant/equivariant algorithms use softmax attention (Alg. 2/3). It’s unclear whether the actual gauge blocks use linear attention kernels (and if so, which feature map) and how gauge invariance is preserved under that kernelization), or only the “feature branch” is linear while the gauge path remains softmax (thus quadratic). Unless I'm missing something this point seems to critically affect the complexity claim.

For DrivAerNet the authors append Euclidean coordinates and normals to the spectral embeddings. This seems to re-introduce a coordinate-system dependence into the very vectors whose inner products are supposed to be gauge-invariant. Please clarify: are coords/normals entering only the values (feature branch) or also the Q/K used for gauge-invariant attention?

The Planetoid/inductive tables omit several post-2022 graph-Transformer baselines that are now standard, e.g. GPS-style models with LapPE/RWPE encodings, sparse-attention Exphormer variants, and/or tokenized/CLS readouts such as NAGphormer/TokenGT. Including these would strengthen empirical positioning.

The text asserts that, in the refinement limit, similarities “recover the continuum Green’s-function kernel” and that self-attention therefore realizes a nonlocal kernel integral. This is interesting, but there is no theorem or convergence experiment (e.g., accuracy vs mesh refinement with fixed parameters) to substantiate discretization invariance beyond heuristic argument.

With a 3-branch block and added geometric channels, it’s unclear which component yields the improvement. An ablation (remove gauge path / remove local path / use softmax vs linear in each) is needed to credit the proposed gauge mechanism vs generic multi-branch modeling and extra features.

Authors should provide wall-clock, peak memory, FLOPs, and parameter counts per branch on Planetoid, PPI, and DrivAerNet; confirm O(N) scaling end-to-end.

**Questions:**

Do the gauge blocks use linear attention or softmax?

If you sample a fresh R' at test time (or per graph), does accuracy remain stable?

Do coordinates/normals ever enter Q/K in the gauge-invariant path? If so, how is invariance preserved?

Operator discretization study: Fix parameters and vary mesh resolution; does error remain flat (up to sampling noise)?

Any empirical support for Green’s-function limit behavior?

---

> ### Author Response · Authors · 2025-11-23
> **Addressing Weaknesses raised by Reviewer Qtqz**
>
> ### Weakness 1: The text claims linear end-to-end via the Katharopoulos et al. linear transformer...
>
> We want to thank the Reviewer for appreciating the conceptual aspects of our work and emphasizing the compelling results on large meshes. We're also glad for the opportunity to clarify our architecture and Alg. 2/3. Indeed, all branches of our multi-scale architecture use linear attention, which as pointed out by the reviewer is crucial for linear scalability.
>
> Importantly, the Gauge-Invariant Spectral self-attention branch (the global branch) implement linear attention with a ReLU feature map, which is very close to the feature map used in Katharopoulos et al. 2020 ($\varphi(x)=elu(x)+1$), but with the added benefit that it has already been proven that it induces a kernel (the arc-cosine kernel as demonstrated by Cho and Saul, 2009). The fact of inducing a kernel guarantees that gauge-invariance is maintained as we explain in an additional section that we included in the revised PDF (from line 294) and in the appendix.
>
> Empirically, the elu(x)+1 feature map (which like relu is positive everywhere) achieves very similar results as relu, and we conjecture that it might also induce a kernel, but leave these types of investigations for future work.
>
> ### Weakness 2: For DrivAerNet the authors append Euclidean coordinates and normals...
>
> The spatial coordinates and normals are processed as usual by the feature branch, and indeed only enter the Gauge-Invariant global branch as value vectors, meaning that they do not affect the gauge-invariant processing of the graph embeddings (see Algorithm 2).
>
> ### Weakness 3: The Planetoid/inductive tables omit several post-2022 graph-Transformer baselines ...
>
> We appreciate the reviewer's comment. We have expanded our experimental comparisons to include additional recent graph Transformer baselines. Specifically, we added rows for SpecFormer, PolyFormer, and Exphormer, which represent current spectral and sparse-attention Transformer variants. These models were selected because they provide relevant architectural parallels and can be feasibly evaluated on our benchmarks. Note that our results are still very competitive with these newer approaches. We did not include GPS-style (LapPE/RWPE), TokenGT, or NAGphormer in the empirical tables, as their full-attention or tokenized formulations exhibit severe memory and scalability limitations on larger graphs (e.g., Arxiv, Elliptic, and PPI), making direct comparison under our inductive settings impractical. Nonetheless, we discuss these models in the related work section and clarify their relationship to our approach.
>
> ### Weakness 4: ... continuum Green’s-function kernel...
>
> We agree with the reviewer about the importance of providing a more formal support to our statement regarding the convergence of Gauge-Invariant Spectral Self-Attention to a Green's function, and the implication for defining a Neural Operator. To this end, we extended the section mentioned by the reviewer with a formal theorem (Proposition 1) with a formal statement of the convergence properties of Gauge-Invariant Spectral Self-Attention, which is proved in the Appendix. In essence, we prove that GIST induces a Neural Operator, since gauge-invariance is a mechanism that allows the meaningful comparison of meshes at different resolutions (which can trivially be interpreted as graphs), and in particular, the error achieved at one discretization level can be used to bound the error at a finer discretization error, and this error goes to zero as the discretizations are refined.
>
> ### Weakness 5: 3-branch block ... ablation ...
>
> Thank you for the suggestion. We added such an ablation study in the Appendix (section "Multi-Scale Architecture Ablation Study"), where we removed individual branches and trained the resulting ablated architectures from scratch. We hope the reviewer will appreciate how this study shows that all three branches meaningfully contribute to the performance on the PPI benchmark (where the full architecture achieves state-of-the-art results), confirming the complementarity of the three branches and providing support for our design choice.
>
> ### Weakness 6: ... confirm O(N) scaling end-to-end
>
> Thank you also for this suggestion. We provide a scaling experiment that confirms the linear scaling of the architecture in the Appendix section "GIST Scalability Study". In particular, we ran a forward and backward pass on DrivAerNet samples while dropping a fraction of mesh nodes and varying that fraction from 100% to 0 in order to simulate input sample graphs of growing size. This study confirms that as the number of input nodes grows from 0 to close to 1M both the VRAM utilization and the time taken to run a forward+backward pass grow linearly.

---

> ### Author Response · Authors · 2025-11-23
> **Answers to questions by Reviewer Qtqz**
>
> 1. *Do the gauge blocks use linear attention or softmax?*
>
>     * All blocks in our architecture use linear attention. This guarantees that our models maintain end-to-end computational complexity scaling that is linear in the input size.
>
> 2. *If you sample a fresh R' at test time (or per graph), does accuracy remain stable?*
>
>     * Yes, this is indeed what happens in inductive tasks (like PPI), where training and test graphs are different and result in independently generated R's.
>
> 3. *Do coordinates/normals ever enter Q/K in the gauge-invariant path? If so, how is invariance preserved?*
>
>     * As alluded to by the reviewer's question and explained above while addressing the raised weaknesses, coordinates and normals do not enter the gauge-invariant path as Q and K vectors, but only as V vectors. Coordinates and normals are in other words treated as features, and other than entering the gauge-invariant branch as V, they enter the feature branch as regular features.
>
> 4. *Operator discretization study: Fix parameters and vary mesh resolution; does error remain flat (up to sampling noise)?*
>
>     * This could be an interesting experiment, but would require creating a bespoke dataset, since for instance DrivAerNet is generated at a fixed discretization. We instead answer this question more in general by providing a formal Theorem (Proposition 1) bounding the discretization mismatch error and explaining how discretization invariance with bounded mismatch error is related to gauge invarianve (in short, graphs corresponding to different discretization can only be meaningfully related if gauge invariance is maintained).
>
> 5. *Any empirical support for Green’s-function limit behavior?*
>
>     * As for the point above, we provide a formal Theorem (as part of Proposition 1) stating the convergence rate of the convergence towards a Green's function and proving that in the Appendix. We hope that the reviewer will be satisfied with us taking a broader more general perspective on the question.

---

### Author Response · Authors · 2025-11-27

Dear Reviewers,

We sincerely appreciate your constructive comments on our paper. We have diligently worked to address your feedback, resulting in what we believe is a significantly improved version. Notably, we have included several additional results, including a formal theorem to support our convergence claims.

For your convenience, we have resubmitted the paper as a PDF with the revisions highlighted, making it easier to locate how we have responded to your concerns.

We look forward to your further feedback and hope for a positive evaluation.

---

> ### Author Response · Authors · 2025-12-01
>
> Dear Reviewers,
>
> We hope you had time to consider our rebuttals. We'd be very grateful if you could comment on whether they satisfactorily addressed your concerns, so as to make it easier for the new AC to properly evaluate our paper in light of the many additions that we introduced to incorporate all comments.
>
> We believe that the new version of the paper that we submitted as part of the rebuttals represents a substantial improvement of our paper, and we hope the reviewers will agree. We rewrote parts of the introduction further clarifying the novelty and significance of our work, and how it connects graph neural networks with neural operators. We included additional experiments, among those: robustness studies on key hyperparameters, ablations studies, and additional GNN benchmarks demonstrating performance comparable to state-of-the-art. We clarified scaling of our algorithms, both formally and empirically with memory and latency scaling experiments. Finally, we included a formal Theorem (Proposition 1) with a related proof in the appendix which theoretically grounds our model as a (graph) Neural Operator with bounded discretization mismatch error.
>
> As said, we believe all these contributions substantially improved on the originally submitted paper, and we thank the reviewers for the suggestions that help that.
>
> Best regards,
> Authors

---

### Meta-Review · Area_Chair_AiUg · 2025-12-23

**Summary:**

This paper proposes GIST, a graph transformer that uses gauge-invariant spectral attention via inner products of randomly projected Laplacian eigenvectors, combined with linear attention for scalability. The reviewers raised several consistent concerns: (1) limited novelty, as the core idea of using distance-based or inner-product operations on spectral embeddings to achieve rotation/sign-flip invariance has precedent in prior work (e.g., "On the Expressive Power of Spectral Invariant Graph Neural Networks," ICML 2024, which explores similar spectral invariants), the twist in this paper of using projections rather than full orthogonal transformations in novel but a bit incremental; (2) insufficient experimental validation, with outdated baselines, missing ablations, and lack of wall-clock/memory scaling experiments in the original submission; (3) clarity issues regarding the use linear attention. While the authors provided substantial revisions, including new baselines, ablation studies, scalability experiments, and a formal proposition, the fundamental concern about limited conceptual novelty remains—the gauge-invariance mechanism, while cleanly presented, represents an incremental contribution.

**Reviewer Concerns:**

Addressed: The authors added requested ablations, scalability experiments, hyperparameter robustness studies, additional baselines, and Proposition 1 formalizing discretization error bounds. Clarifications about linear attention usage and coordinate handling were provided.
Outstanding: The core novelty concern persists—inner-product-based gauge invariance is not substantially different from distance-based spectral invariants in prior work. The benchmarks remain underwhelming.

**Reviewer Scores:**

Even under the most optimistic interpretation of the rebuttal's impact, where each reviewer raises their score by two points, the resulting scores (approximately 6, 4, 6) would still place this submission in borderline territory. Given the outstanding concerns regarding novelty relative to existing spectral invariant methods and the incremental nature of the contribution, I would not recommend acceptance even in this best-case scenario.

---

### Decision · Program_Chairs · 2026-01-26

Reject